



# Assessing hydrological sensitivity of grassland basins in the Canadian Prairies to climate using a basin classification–based virtual modelling approach

Christopher Spence[1*], Zhihua He[2], Kevin R. Shook[2], Balew A. Mekonnen[3], John W. Pomeroy[2],
Colin J. Whitfield[4], Jared D. Wolfe[5]

* Corresponding author: Christopher Spence (chris.spence@canada.ca)

[1] Environment and Climate Change Canada, Saskatoon, Saskatchewan, Canada

[2] Centre for Hydrology, University of Saskatchewan, Saskatoon, Saskatchewan, Canada

[3] Golder Associates, Calgary, Alberta, Canada

[4] School of Environment and Sustainability, University of Saskatchewan, Saskatoon, Saskatchewan, Canada

[5] Saskatchewan Ministry of Environment, Regina, Saskatchewan, Canada

*corresponding author: chris.spence@canada.ca

**Abstract**

Significant challenges from changes in climate and land-use face sustainable water use in the Canadian Prairies ecozone. The region has experienced significant warming since the mid 20th Century, and continued warming of an additional 2°C by 2050 is expected. This paper aims to enhance understanding of climate controls on Prairie basin hydrology through numerical model experiments. It approaches this by developing a basin classification–based virtual modeling framework for a portion of the Prairie region, and applying the modelling framework to investigate the hydrological sensitivity of one Prairie basin class (High Elevation Grasslands) to changes in climate. High Elevation Grasslands dominate much of central and southern Alberta and parts of southwestern Saskatchewan with outliers in eastern Saskatchewan and western Manitoba. The experiments revealed that High Elevation Grasslands snowpacks are highly sensitive to changes in climate, but that this varies geographically. Spring maximum snow water equivalent in grasslands decreases 8% per degree °C of warming. Climate scenario simulations indicated a 2°C increase in temperature requires at least an increase of 20% in mean annual precipitation for there to be enough additional snowfall to compensate for enhanced melt losses. The sensitivity in runoff is less linear and varies substantially across the study domain; simulations using 6°C of warming and a 30% increase in mean annual precipitation yields simulated decreases in annual runoff of 40% in climates of the western Prairie but 55% increases in climates of eastern portions. These results can be used to identify those areas of the region that are most sensitive to climate change, and highlight focus areas for monitoring and adaptation. The results also demonstrate how a basin classification–based virtual modeling framework can be applied to evaluate regional scale impacts of climate change with relatively high spatial resolution, in a robust, effective and efficient manner.

**Key words:** Prairie, basin classification, virtual experiments, climate change, snow, runoff



## Introduction

Hydrological models are essential tools to understand hydrological processes and function at the
basin scale, and can also be used to diagnose how specific hydrological processes control

catchment responses to change (Rasouli et al., 2014). Modelling a specific basin to evaluate
processes or to simulate the effects of change entails large computational and labour costs and
requires observations of the basin response with sufficient spatial and temporal coverage.
Modelling of many individual basins is not efficient when attempting to predict regional
responses to changes in climate and/or land-use. Basin classification can regionalize

hydrological model outputs, based on the assumption that basins can be classified by their
characteristics and that basins of the same class respond similarly to changes in climate inputs or
their landscapes (e.g., McDonnell and Woods, 2004; Wagener et al., 2007). Parameterizing a
model based upon a representative or stylized basin of a given class allows the output to be
considered representative of all basins of that class. This assumption facilitates regionalization

as it does not necessarily require simulating the distinctive characteristics of every basin,
reducing cost and time required for large domain studies. Such a regionalization approach can
be used to assess the sensitivity of large diverse areas to stressors, such as land-use and climate
change.

One such region is the Canadian Prairie ecozone, that portion of the Great Plains of North
America that includes southern parts of the provinces of Alberta, Saskatchewan, and Manitoba
and Treaties 1, 2, 4, 6 and 7 in Western Canada (Spence et al., 2019), as mapped in Figure 2.
This region has a cold sub-humid to semi-arid climate and was covered by grassland and sparse
woodlands until the widespread adoption of cultivated agriculture in the late 19[th] and early 20[th]



centuries. The region's geomorphology was formed by glacial and post-glacial processes which

left numerous internally drained depressions and poorly defined drainage networks. Most of the

Canadian Prairies is in the Saskatchewan-Nelson River Basin, but relatively little runoff is

provided to the major rivers that traverse the region downstream of their mountain headwaters.

Local streams and prairie-derived rivers often have intermittent and highly variable streamflow.

These streams are important local sources of freshwater and are often managed to provide farm,

agricultural and municipal water supply and support natural lakes and reservoirs (Pomeroy et al.,

2005). Because they connect to larger systems only intermittently, a small headwater–basin scale

approach is necessary to generate information about how their behaviour might be impacted by

the aforementioned stressors.


Western Canada, including the Canadian Prairies, has been subject to substantial climate

warming since the mid 20[th] century (DeBeer et al., 2016; Bush and Lemmen, 2019). Prairie

precipitation trends indicate more rain and less snow in the spring and fall (Shook and Pomeroy,

2012) and runoff generation has been shown to be shifting from snowmelt- to rainfall-driven in

eastern Saskatchewan (Dumanski et al., 2015). Recent analysis of hydrometric stations across the

region identified sub-regional trends in streamflow associated with drying in the west and south

and wetting in the east and north, associated with physical landscape characteristics and climate

(Whitfield et al. 2020). However, it is difficult to attribute streamflow response solely to climate

change because of impoundment of streams, widespread changes in agricultural practices and

wetland drainage since the 1950s (Ehsanzadeh, 2016). Wetland drainage has become

widespread in portions of the region (van Meter and Basu, 2015) and the loss of depressional

storage capacity associated with drainage enhances streamflow volumes (Tiner, 2003; Fang et



al., 2010; Wilson et al., 2019) and may alter the frequency, timing, and duration of regional

streamflow (Ehsanzadeh et al., 2012; Spence and Mengistu, 2019). Extrapolating intensive

studies of wetland drainage impact in individual basins (Wilson et al., 2019) can be challenging,

because basin response is a function of wetland distributions that control contributing area

dynamics (Stichling and Blackwell, 1957; Shaw et al., 2012; Shook and Pomeroy, 2011; Haque

et al. 2017, Spence and Mengistu, 2019). It is uncertain how hydrological fluxes and states in

Canadian Prairie basins will respond to continued climate change and wetland drainage. The

statistical modelling and small basin modelling studies cited here have provided an excellent

foundation, but an improved approach is needed to evaluate how changes in climate and

agricultural practices impact hydrological regimes more broadly across the region.

Here, a classification-based virtual modeling framework is proposed as a means to examine

hydrological sensitivity to different climate, land-use and wetland drainage. In this approach,

each basin class is modelled in a virtual manner (Weiler and McDonnell, 2004; Armstrong et al.,

2015); as a synthetic or generic basin with characteristics defined by the average or typical

condition of all basins from the same class. In this way, the basin characteristics can be

manipulated to determine how a typical basin may respond to change. There is evidence that

such an approach is viable, as virtual experiments have been used to evaluate hydrological

response to different conditions (Di Giammarco et al., 1996; Horn et al., 2005; Dunn et al., 2007;

Mallard et al., 2014; Seo and Schmidt, 2013, Lopez-Moreno et al., 2020), identify factors

influencing hydrological processes (e.g., Weiler and McDonnell, 2004), and study hydrological

controls on water chemistry (Weiler and McDonnell, 2006). This paper aims to demonstrate the

utility of a basin classification–based virtual modelling approach for assessing the sensitivity of

Canadian Prairie catchments to climate. Two steps were taken to achieve this objective: (1)

development of a robust class-based virtual basin model for a portion of the Canadian Prairie

and; (2) exploration of virtual basin sensitivity of hydrological response to climate. This work

provides a foundation to extend the virtual basin modelling approach more broadly across the

Canadian Prairie to assess response to climate and land management scenarios.

**Methodology**

*Framework of classification-based virtual basin modeling*

A basin classification–based virtual modelling platform has three main components: (1) a

classification analysis to derive virtual basin characteristics; (2) parameterization and evaluation

of a hydrological model of the virtual basin and (3) application of the model to evaluate response

to multiple scenarios (Figure 1).

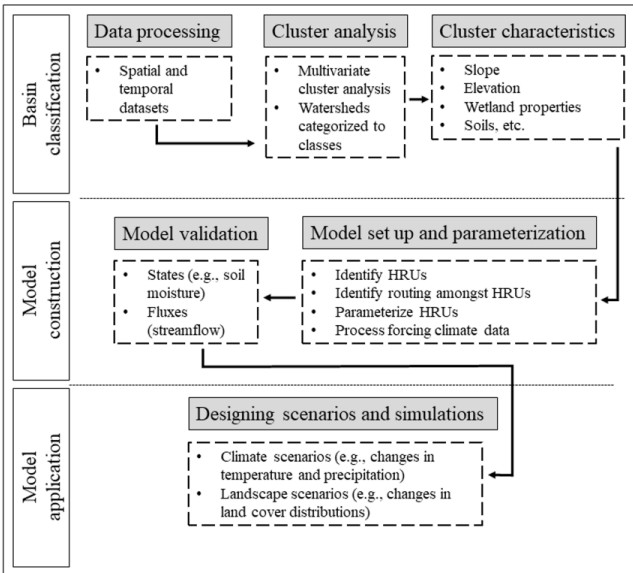

Figure 1: Components of the classification-based virtual basin modeling platform.




*Basin Classification*

The classification of Canadian Prairie basins was based on the analyses of Wolfe et al. (2019), which divided over 4000 basins, each approximately 100 km$^2$ in area, into seven broad classes, based on a suite of physio-geographic characteristics (Figure 2). The basin delineations used in the study were taken from the HydroSHEDs dataset (Lehner and Grill, 2013), which provides geographically contiguous delineations of basins for the world including the Prairie ecozone. Physio-geographic characteristics, including climate, geology, topography, wetland distribution, and land cover, among others, were compiled and used to classify basins that would be expected to respond in a hydrologically coherent manner (Wolfe et al. 2019). Urban areas and large lakes were excluded from the analysis. A revised classification that excluded climate parameters was conducted and is used herein, following the same Hierarchical Classification of Principal Components (HCPC) approach. This was done because climate is introduced through the long meteorological time series used to drive the virtual basin model and in order to study climate sensitivity any classification that included historical climates could introduce bias. Exclusion of climate had a limited impact on the basin classification, with the seven classes of basins identified (Figure 2) closely following the original classification.

The High Elevation Grasslands (HEG) class (Figure 2) was selected for the development of the virtual basin model. This class featured 751 basins with an average size of ~100 km$^2$. Median basin characteristics, including land cover fractions, basin slope and elevation, and soil type, were determined and used for virtual basin model parameterization. Wetland areas derived from the Global Surface Water maximum water extent dataset (https://global-surface-water.appspot.com/download) were used to determine the shape and scale parameters of a





generalized Pareto distribution (Shook et al. 2013, Table 1). These parameters were used to

characterize the wetland complex within the virtual model, where the wetland complex is

represented by multiple individual wetlands of varying sizes.

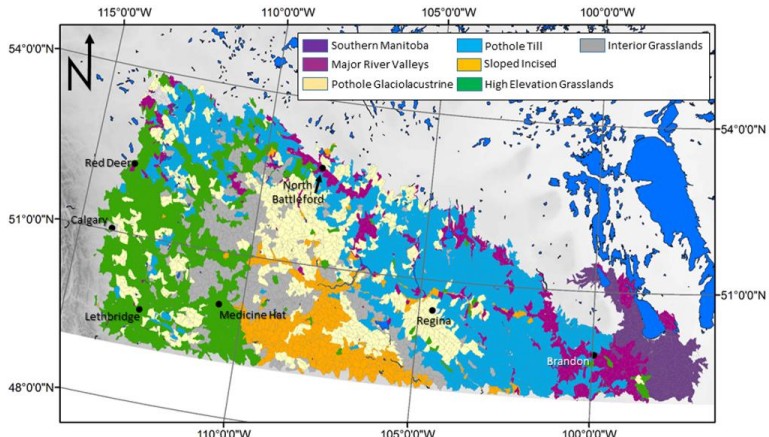

Figure 2: Basin classification map for the Prairie ecozone in western Canada. Colours represent
the seven basin classes and excluded areas (lakes and urban areas) are shown in white.  The
focus of this paper is on the High Elevation Grasslands, shown in green. Climate data from the
cities noted in this figure were used to drive the virtual model. Boundaries of the basins used in
the classification are shown in grey

Table 1:  CRHM parameters for the High Elevation Grassland virtual basin model. The suffix "-
w" in the HRU name indicates HRUs in the wetland catena. LAI denotes leaf area index.

| HRU | Fraction of basin | LAI | Fetch (m) | Vegetation height (m) | Stalk density (/m²) | Stalk diameter (m) | Manning's n |
|---|---|---|---|---|---|---|---|
| Channel (CH) | 0.01 | 0.001 | 300 | 0.50 | 1 | 0.003 | 0.07 |
| Cultivated (CL) | 0.32 | 0.001 | 1000 | 0.20 | 320 | 0.003 | 0.17 |
| Cultivated (CL-w) | 0.13 | 0.001 | 1000 | 0.20 | 320 | 0.003 | 0.17 |
| Fallow (FL) | 0.004 | 0.001 | 1000 | 0.01 | 320 | 0.003 | 0.05 |
| Fallow (FL-w) | 0.002 | 0.001 | 1000 | 0.01 | 320 | 0.003 | 0.05 |
| Grassland (GL) | 0.30 | 0.001 | 500 | 0.40 | 320 | 0.003 | 0.2 |
| Grassland (GL-w) | 0.12 | 0.001 | 500 | 0.40 | 320 | 0.003 | 0.2 |
| Shrubland (SL) | 0.02 | 0.001 | 300 | 1.50 | 100 | 0.01 | 0.2 |
| Shrubland (SL-w) | 0.006 | 0.001 | 300 | 1.50 | 100 | 0.01 | 0.2 |





| | | | | | | | |
|---|---|---|---|---|---|---|---|
| Woodland (WL) | 0.02 | 0.4 | 300 | 6.00 | 100 | 0.1 | 0.4 |
| Woodland (WL-w) | 0.006 | 0.4 | 300 | 6.00 | 100 | 0.1 | 0.4 |
| Wetland (WT) | 0.07 | 0.001 | 300 | 1.50 | 1 | 0.01 | 0.2 |

**Albedo**

| | |
|---|---|
| Bare ground | 0.17 |
| Snow (fresh) | 0.85 |

**Wetland area Generalized Pareto Distribution parameters**

| | |
|---|---|
| Shape ($\xi$) | 1.152 |
| Scale ($\beta$) | 2070.62 |

*Model set-up and parameterization*

The Cold Regions Hydrological Modelling platform (CRHM) was selected to develop the virtual

basin model, as CRHM is particularly suited for simulating the hydrology of the Canadian

Prairies.  CRHM is a modular, process-based, spatially semi-distributed hydrological model,

which includes the key cold regions and warm season hydrological processes that operate in

western Canada and elsewhere (Pomeroy et al., 2007). With the correct suite of modules, each

representing a key hydrological process, CRHM has proven very capable of representing prairie

hydrological processes and accurately emulating water fluxes in this landscape (Fang and

Pomeroy, 2009; Fang et al., 2010).  Most of the process modules in CRHM, particularly the

surface processes, are strongly physically-based, and hence CRHM does not require model

calibration, making it suitable for simulations under nonstationary conditions.  As a virtual basin

has no specific location, it cannot be calibrated to field observations from a particular basin, so

setting parameters based on regional hydrological studies rather than calibration is advantageous.

The virtual basin area was set to 100 km$^2$, which aligns with the average basin size used in the

classification. The virtual basin was divided into hydrological response units (HRUs), each of

which has a single set of parameter values and for which water budgets are calculated.  The HRU





sequence by which runoff is routed follows a catena of land cover from cultivated fields at the

highest elevation, followed by grasslands, shrublands, woodlands, and wetlands at successively

lower elevations. Areas were set according to the median for that land cover observed across all

HEG basins. Summer-fallow fields, whilst also included, cover a very small area (<1%; Table 1).


The virtual basin was separated into non-wetland, and wetland catenas (Figure 3) according to

effective and non-effective fractions of HEG basins, respectively. The first, 'non-wetland' catena

portion (i.e., cultivated, grassland, shrubland, woodland HRUs) of the basin (~67% of area) is

considered to contribute flows directly to the HRU outlet (stream channel). Runoff from the

'wetland' catena portion of the virtual basin (~33% of area) features a wetland complex HRU

within a landscape catena following a sequence from cultivated, to grassland, shrubland, and

woodland HRUs (Figure 3). Runoff is routed through a wetland complex comprised of 46

individual wetlands; their areas follow a generalized Pareto distribution (Shook et al. 2013). This

approach has been shown to effectively represent how wetlands dictate transmission of runoff

from Prairie basins, by representing the dynamic area contributing flow downstream (Pomeroy et

al., 2014).





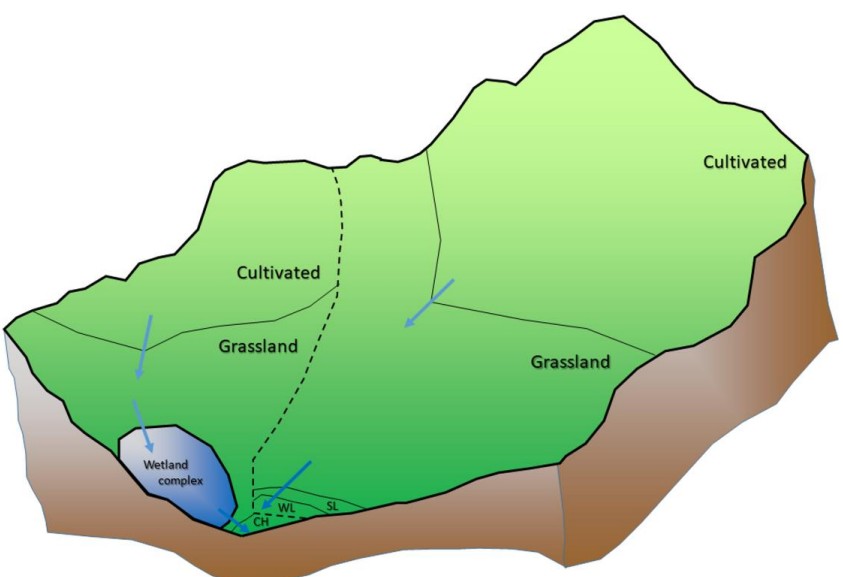

Figure 3: Illustration of HRU distribution in the virtual model of a typical High Elevation
Grasslands basin. The illustration depicts two catenas, one with a wetland complex and one
without, separated by the dashed line. Relative areas are meant to approximate those listed in
Table 1. SL denotes shrublands; WL, woodlands; and CH a channel HRU (see Table 1). The
fallow and shrubland and woodland HRU's directed through the wetland complex are too small
to be shown at this scale.

The Prairie Hydrological Model (PHM), created using CRHM (Pomeroy et al., 2010, 2012), was

used to simulate virtual basin hydrological response. PHM includes a specific set of physically-

based modules linked sequentially to represent the dominant hydrological processes for the

virtual basin. The Observation module reads in meteorological data and calculates the

precipitation phase using a maximum temperature threshold of 0°C for snow and a minimum

temperature threshold of +2°C and distributes these to each HRU along with lapse rates for

temperature and precipitation and temporal interpolation to adjust observation time steps to

hourly data. The Radiation module outputs incoming shortwave radiation to slopes, longwave

radiation, and net radiation. The Canopy module (Sicart et al., 2004; Ellis et al., 2010) was used

to adjust the effects of vegetation canopies on sub-canopy net radiation reaching underlying



snow.  The leaf area index was assigned a value of 0.4 for woodland HRUs, which is a typical

value for aspen trees during winter (Pomeroy et al., 1999). For other HRUs, a minimum value of

0.001 was set to simulate the canopy effects of Prairie vegetation (crop residue, grass) on

radiation for snowmelt. The values of the initial albedos for bare ground and fresh snow were set

to 0.17 and 0.85, respectively (Table 1), based on suggested values by Armstrong et al. (2008)

for summer conditions and Male and Gray (1981) for snow.

To simulate snow redistribution processes, the Prairie Blowing Snow Model (PBSM) (Pomeroy

and Li, 2000), and Walmsley's windflow model (Walmsley et al., 1989) modules were used.

Fetch distances were set using values recommended in Pomeroy et al. (2007). Fetch distance was

set to 1000 m for exposed sites such as cultivated and fallow HRUs. A 500 m fetch was assigned

for grassland HRUs. For other HRUs, a 300 m fetch was selected.  The distribution factor

parameterizes the allocation of blowing snow transport from aerodynamically smoother (or

windier) HRU to aerodynamically rougher (or calmer) ones and was selected according to the

prairie landscape aerodynamic sequencing of Fang and Pomeroy (2009). Vegetation height, stalk

density, and stalk diameter were set to represent the values for the prairie environment during fall

and winter (Table 1) (Pomeroy and Li, 2000).

The Energy-Budget Snowmelt Model (EBSM) (Gray and Landine, 1988) module includes

algorithms applicable to the Canadian Prairies and was used to simulate snowmelt by calculating

the balance of radiation, sensible heat, latent heat, ground heat, advection from rainfall, and

change in internal energy. Infiltration to unfrozen and frozen soils was calculated by the Prairie

Infiltration module with algorithms based on Ayers (1959) and Gray et al. (1985), respectively.



Actual evapotranspiration was simulated using the Penman-Monteith equation (Monteith, 1965),

and evaporation from typically saturated surfaces subject to advection, such as wetlands and

stream channels, was calculated using the Priestley and Taylor equation (Priestley and Taylor,

1972).

The Soil module calculates the water balance for the soil column, which is divided into two

layers: the top layer (called the recharge zone) and the lower layer. The depth of the recharge

layer was set at 1.4 m as this is typically a zone of higher hydraulic conductivity in the glacial till

derived soils of the region (Brannen et al., 2015). Wolfe et al. (2019) identified the predominant

HEG soil texture as loam, which was then assigned as the soil type for all HRUs. The

Muskingum routing method was used for the routing amongst HRUs. The routing length for each

HRU was calculated using the modified Hack's Law length-area relationship, which was derived

from a previous CRHM-PHM modeling study of Smith Creek in Saskatchewan (Fang et al.,

2010; Pomeroy et al., 2010).

*Model application*

To ensure that the role of climate variability across HEG was captured in streamflow simulations

the model was run over a 46-year baseline period (1960–2006) driven using data from seven

locations.  The locations were within and nearby the geographical extent of the HEG

classification, and represented the variation in climate across the region (Figure 2; Table 2). The

mean annual average temperature at these seven sites ranged from +2.1°C to +6.1°C and mean

annual precipitation ranged from 323 to 487 mm (Table 2). The virtual basin model was run

using daily precipitation data from the Adjusted and Homogenized Canadian Climate Data





(AHCCD) (Mekis and Vincent, 2011; Vincent et al., 2012) collected at these seven locations.

This dataset corrects shifts identified due to station relocation and changes in observing practices

and automation. Other discontinuities are adjusted with multiple linear regression using a

penalized maximal t-test and a quantile-matching algorithm. For precipitation, corrections are

applied to account for wind undercatch, evaporation, and gauge-specific wetting losses. Snowfall

density corrections are derived based on coincident ruler and Nipher measurements. Trace

precipitation is added. The daily precipitation data were converted to hourly data required by

CRHM using linear interpolation using the Observation module. The other hourly forcing

variables (temperature, relative humidity and wind speed) were taken from Environment and

Climate Change Canada observations for stations for the same seven locations. The data were

quality controlled and infilled using nearby station data.

Table 2: Climate characteristics (1981–2010 climate normal) of the seven selected locations
located in and near the High Elevation Grassland class. $T_a$ is mean annual temperature and P
denotes mean annual precipitation

| Location | Latitude | Longitude | $T_a$ (°C) | P (mm) |
|---|---|---|---|---|
| Red Deer, Alberta | 52° 16' 22" N | 113° 48' 36" W | 2.8 | 487 |
| Calgary, Alberta | 51° 02' 37" N | 114° 04' 33" W | 4.4 | 419 |
| Medicine Hat, Alberta | 50° 02' 43" N | 110° 40' 05" W | 6.1 | 323 |
| Lethbridge, Alberta | 49° 41' 32" N | 112° 51' 07" W | 5.9 | 380 |
| North Battleford, Saskatchewan | 52° 46' 37" N | 108° 17' 43" W | 2.1 | 374 |
| Regina, Saskatchewan | 50° 26' 58" N | 104° 37' 10" W | 3.1 | 390 |
| Brandon, Manitoba | 49° 50' 51" N | 99° 56' 50" W | 2.2 | 474 |

Outputs from the baseline simulations (1965–2006) were used to assess whether the virtual basin

model captured typical behaviour of HEG basins. The first five years of simulations were

excluded from the analysis to avoid the potential to misrepresent initial conditions due to lag

effects associated with antecedent conditions. Because the virtual basin is designed to represent



typical behaviour of a basin class, it does not reproduce the hydrology of any specific basin and there is no direct physical analogue for which observations can be used to completely assess

performance. Previous studies have described the application of CRHM to Canadian Prairie basins, and its ability to represent the region's predominant hydrological processes is well established (Fang et al., 2010). These findings lend confidence that the virtual basin model, informed by these other CRHM applications, can be applied for the diagnostic purposes of this study. Furthermore, the aim of the simulations was not to simulate specific basins in the region,

but to assess the sensitivity of the hydrological regime to changes in climate forcings. For this reason, spring snow water equivalent (SWE) values from snow courses and mean annual hydrographs from hydrometric gauges at multiple sites within the HEG class were compared to virtual basin model outputs to establish that the virtual basin model was capturing the correct timing and magnitudes of important states and fluxes. Runoff ratios were also compared to

establish if the virtual model was dividing the water budget in a reasonable manner.

To evaluate spring snow accumulation, measured SWE at the Alberta Environment and Parks Wetaskiwin, Kneehill Valley and Innisfail East snow courses (Table 3) were compared against modelled SWE values for the cultivated HRU on the same day measurements were taken, using

climate data from the nearest station (Red Deer) (Table 2), for the period during which there was data overlap (1987–2006). It is recognized that snow accumulation in this region is not continuous throughout the winter as there can be significant ablation events, however, measurements are not taken earlier than late February. Mean monthly discharge depths from basins 100% within the HEG class were generated using climate data from four nearby

meteorological stations (Table 2) and were plotted and visually compared to the 14 Water Survey





of Canada hydrometric stations gauging a stream within 100 km of one of the meteorological

stations (Table 3). The selected basins and meteorological stations used for these evaluations

were all in Alberta as this was where the class was most common and contiguous. These tests

were to discern if the virtual basin model was capturing streamflow seasonality, variability and

annual runoff ratios.

Table 3: Sources of observed data for model evaluation. Water Survey of Canada's (WSC) hydrometric data were obtained from the HYDAT database, available at https://collaboration.cmc.ec.gc.ca/cmc/hydrometrics/www/. Effective drainage area is defined
by Godwin and Martin (1975) as the drainage area that contributes streamflow to the gauged location during the median annual flood was also obtained from HYDAT. The snow water equivalent data were obtained from Alberta Environment and Parks snow courses.

| Station name | Period of record used for validation | Associated climate location(s) | Gross drainage area ($km^2$) | Effective drainage area ($km^2$) |
|---|---|---|---|---|
| **Hydrometric** | | | | |
| Battle River near Ponoka (05FA001) | 1976 - 2006 | Red Deer | 1820 | 1550 |
| Maskwa Creek No. 1 above Bearhills Lake (05FA014) | 1976 - 2006 | Red Deer / Calgary | 79.1 | 61.2 |
| Renwick Creek near Three Hills (05CE011) | 1976 - 2006 | Red Deer / Calgary | 58.9 | 58.1 |
| Ray Creek near Innisfail (05CE010) | 1976 - 2006 | Red Deer | 44.4 | 44.4 |
| Blindman River near Blackfalds (05CC001) | 1976 - 2006 | Red Deer | 1796 | 1460 |
| Battle Creek at Alberta Boundary (11AB117) | 1976 - 2006 | Medicine Hat | 111 | 111 |
| Gros Ventre Creek near Dunmore (05AH037) | 1976 - 2006 | Medicine Hat | 215 | 206 |
| Beaver Creek near Brocket (05AB013) | 1976 - 2006 | Lethbridge | 257 | 257 |
| Prairie Blood Coulee near Lethbridge (05AD035) | 1976 – 2006 | Lethbridge | 214 | 214 |
| Pothole Creek near Magrath (05AE011) | 1976 – 2006 | Lethbridge | 372 | 351 |
| Snake Creek near Vulcan (05AC030) | 1976 – 2006 | Lethbridge | 350 | 346 |





| Trout Creek near Granum (05AB005) | 1976 – 2006 | Lethbridge | 441 | 441 |
| West Arrowwood Creek near Ensign (05BM018) | 1976 – 2006 | Calgary | 30 | 30 |
| **Snow Course** | | | | |
| Innisfail East (05CE801) | 1987 - 2006 | Red Deer | | |
| Wetaskiwin (05FA801) | 1987 - 2006 | Red Deer | | |
| Kneehill Valley | 1987 - 2006 | Red Deer | | |


After evaluation, the virtual basin model was used to explore (1) the role of climate variability

within the HEG class, and (2) how hydrological patterns may change in response to potential

future shifts in climate. Future climate scenarios were investigated using the delta method, which

applies uniform changes to historic daily air temperature and total precipitation records (Lopez-

Moreno et al., 2012; Rasouli et al., 2014). This method has the advantages of being

computationally inexpensive, while avoiding bias, and preserving the covariances among

variables, which are important in modeling cold-regions processes (Shook and Pomeroy, 2010).

It has the disadvantage of not assessing sensitivity to changing extreme events, and does not

account for seasonal differences associated with climate change. Temperature increases of up to

+6°C and precipitation decreases of 20% and increases of up to 30%, based on model projections

for southern Alberta by 2050 (Zhang et al. 2018) were investigated. Within these ranges,

multiple scenarios were run using increments of $1^{\circ}$C (temperature) and 10% (precipitation),

totalling 35 scenarios for each of the seven climate locations. These scenarios were used with the

model to quantify sensitivity of snow accumulation and annual runoff to climate change. The

spatial distributions of the sensitivities were mapped by extrapolating the values for these seven

model locations (Table 2, Figure 2) in ArcGIS using ordinary kriging, assuming a spherical

semivariogram and a lag value of 0.01964.





**Results**

The HEG class occupies much of the western portion of the Canadian Prairies, and includes the

majority of southern Alberta and several isolated patches in both Saskatchewan and Manitoba

(Figure 2). Basins in this class tend to have a high fraction of native grasslands (40% of the

basin area). The elevations of these basins are amongst the highest in the Canadian Prairies. The

mean wetland density of 0.8 km$^{-2}$ is amongst the smallest of the virtual basin classes (Wolfe et

al., 2019). Relatively dense drainage networks, coupled with the small wetland densities, result

in 67% of the gross drainage area of a typical HEG basin contributing to runoff to the outlet in a

median flow year. This is a relatively high percentage for a Canadian Prairie basin.

*Virtual basin model performance*

*Spring snow water equivalent*

The virtual model captured the intra- and inter- annual variability in snow water equivalent

(SWE) quite well as indicated by comparison of CRHM HEG model simulated SWE (Red Deer

forcing data) and snow survey course data (Figure 4). The simulated values were calculated on

the same days as the snow surveys with dates varying between February and April. The

agreement is quite good considering that a) the forcing meteorological data were collected up to

60 km from the snow survey site, and b) that a virtual model, not tuned to the specific traits of

the snow survey location (e.g., vegetation height), was used. Simulated and observed means and

standard deviations were similar, but in two instances the model simulated SWE values notably

larger than observed during the comparison period (Figure 4). A best fit line with a slope of 1.1

and r$^2$ of 0.53 can be drawn among the simulated and observed data.



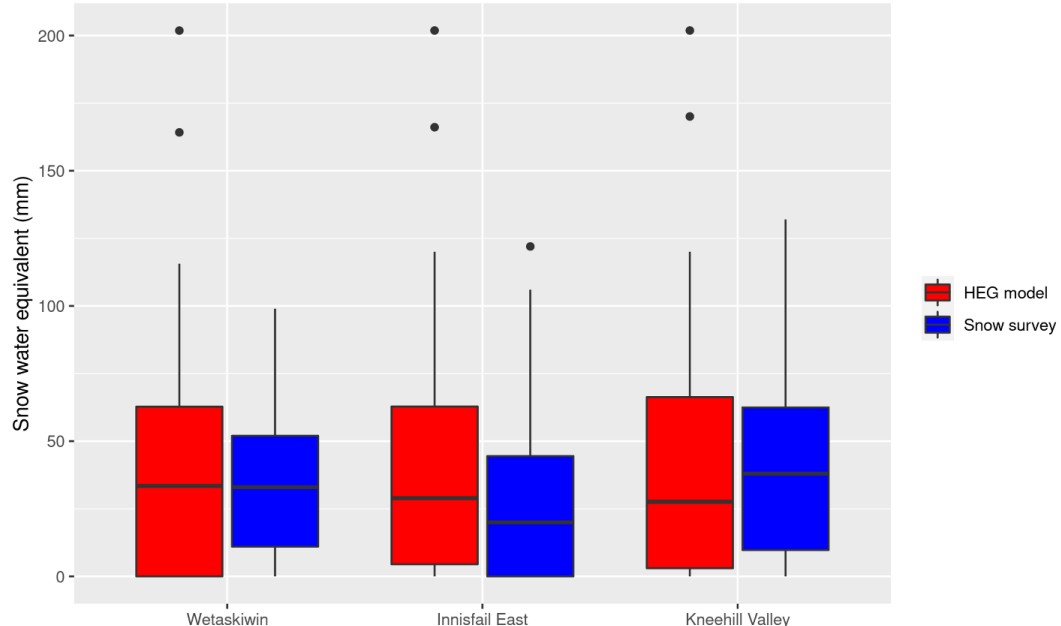

Figure 4: Comparison of 1987–2006 simulated SWE using the Red Deer climate forcings and SWE observed at the three Alberta Environment and Parks (AEP) snow courses near Red Deer.

*Mean annual hydrographs and runoff ratios*

As with SWE, the CRHM HEG model produces reasonable simulations of streamflow. Monthly discharge depths for the HEG model driven by climate forcings from the four Albertan climate locations were in good agreement with WSC gauged streams in the HEG classification (Figure 5). The monthly means were computed only from those years for which data were available for

the HEG virtual model and for all gauging sites. As would be expected, the largest median monthly discharges are found in March through May, due to the spring melt of the accumulated snowpacks. The HEG virtual model did a good job of reproducing the timing and magnitudes of monthly discharge depths, particularly for March and April. This is impressive considering that the meteorological data were collected up to 60 km away from the stream gauging sites. The





agreement between the simulated and gauged discharge depths was poorer in the summer,

particularly in June and July. Mean conditions are well simulated, but there are numerous

instances of outliers in the observed time series, especially in May and June (e.g., Lethbridge –

Figure 5) not captured by the simulations. These events are undoubtedly the result of small, fast-

moving, short duration and intense convective storms rainfall events. The use of daily

precipitation reduces rainfall intensities in the meteorological forcings, and appear less in the

simulated time series.

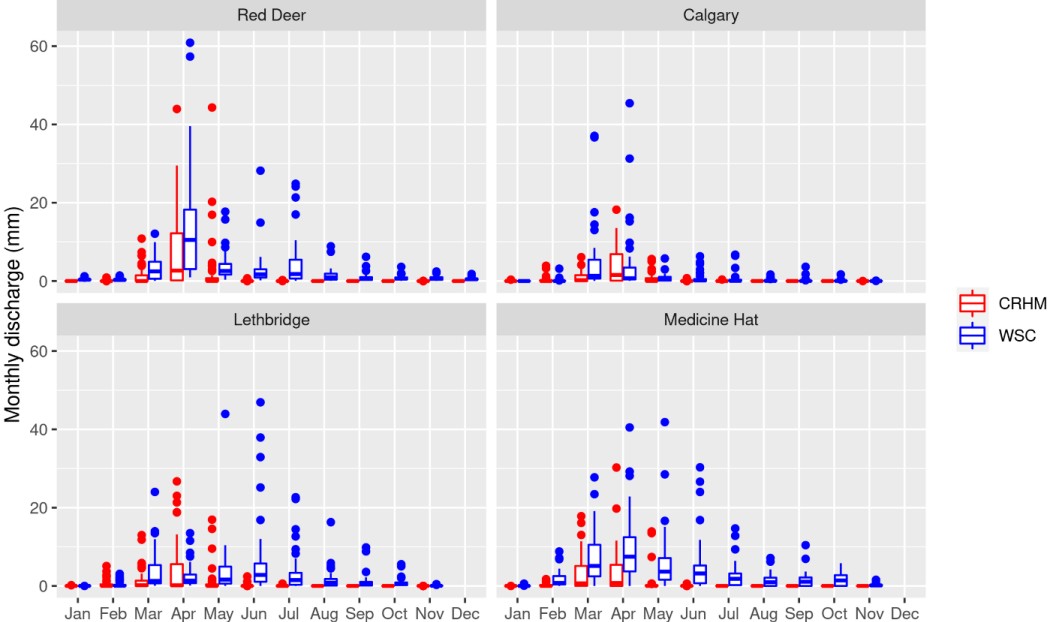

Figure 5: Simulated (CRHM) monthly simulated discharge depths (red) using forcing from four
of the climate locations compared with those from WSC gauged streams (blue). The horizontal
line in the middle of the box denotes the mean, and the top and bottom of the box denote plus or
minus one standard deviation. The whiskers denote 10% and 90% percentiles. Circles represent
values beyond these percentiles.

Runoff ratios were evaluated to determine if the virtual model was dividing the water budget

correctly and producing a reasonable amount of annual runoff. The mean simulated annual



runoff ratios for HEG basins during 1965-2006 were 0.08 ± 0.11. These values are slightly lower than those from the literature. Most documented values of runoff ratios from this landscape are from hillslope and agricultural field scales. Woo and Rowsell (1993) estimated the mean value of annual runoff ratios from a grassland slope in Saskatchewan to be 0.13, which is within the range estimated by the model. Rainfall runoff ratios never exceeded 0.11 from grassland slopes observed by Neath and Chanasyk (1996) in the fescue grasslands of southern Alberta, which aligns well with the lower streamflows simulated by the model after spring melt (Figure 5). Pavlovskii et al. (2019) observed hillslope scale runoff ratios for snowmelt events that have a wide range of values, with lower values experienced in mid-winter (~0.3) (i.e., January and February) versus those in spring (~0.5). Higher runoff is often documented when using Water Survey of Canada gauge data than from the virtual basin model (Whitfield et al., 2020), partly because most basins gauged by the Water Survey of Canada have median contributing area fractions that approach 1.0, higher than the typical value for HEG basins, and are more efficient at producing streamflow.

*Hydrological sensitivity to climate*

*Snow cover*

According to Li et al. (2019) and Zhang et al. (2018) future climates at the end of the 21st century for the region could be as much as 30% wetter and 6°C warmer than the mid 20th century. The effects of these changes on snow cover differ among HRUs (Figure 6; Table 4). The virtual basin model results support the well-documented cold regions phenomenon in that the annual peak SWE declines with warming (Najafi et al., 2017; Fang and Pomeroy, 2008). Source HRUs for redistribution of snow (e.g., cultivated HRUs) experience small absolute changes. This may





be because blowing snow is affected by vegetation height, and a 6°C temperature rise does not

reduce the depth of accumulated snow below this threshold. HRUs receiving blowing snow,

such as wetlands, are more sensitive to warming (Figure 6). Warming of $6^{\circ}$C with no change in

mean annual precipitation decreased the simulated annual peak SWE by 27% in the cultivated

HRU and 51% in the wetland HRU in a climate comparable to Medicine Hat's– this is similar to

the sensitivity of snowdrift SWE to warming found by Fang and Pomeroy (2007; 2009) and is

due to the temperature sensitivity of the occurrence of blowing snow (Li and Pomeroy, 1997).

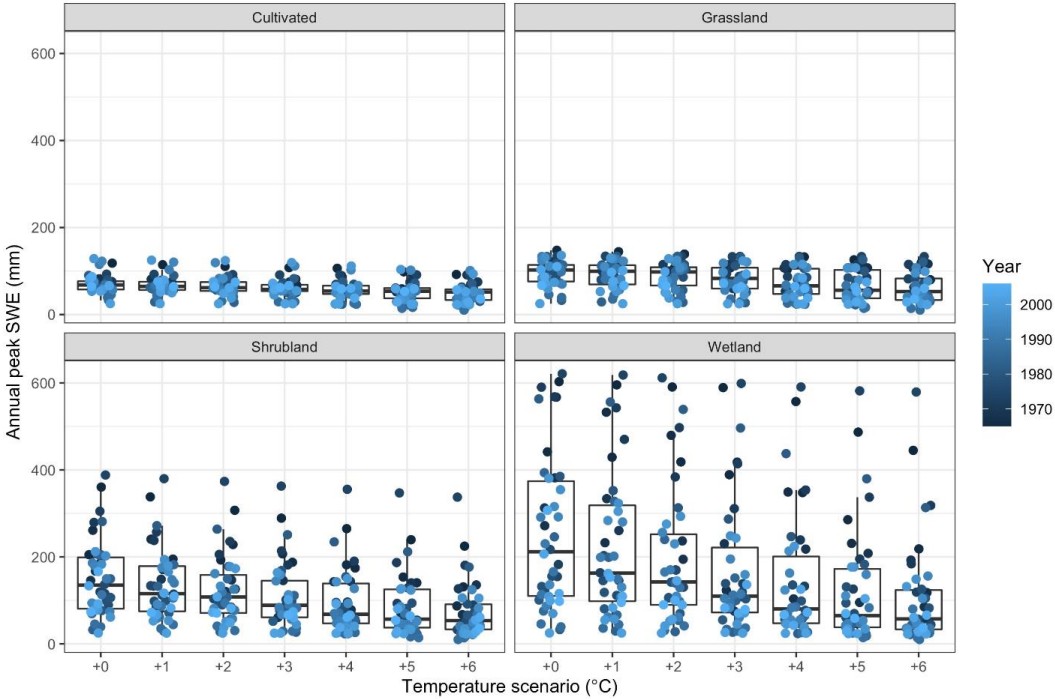

Figure 6: Peak virtual model simulated annual snow accumulation (SWE) for cultivated,
grassland, shrubland, and wetland HRUs for Medicine Hat climate under baseline conditions
(1965–2006: $0^{\circ}$C) and warming of air temperature up to $6^{\circ}$C.





Table 4: Simulated annual peak SWE and runoff under different climate scenarios (temperature
and precipitation expressed as °C or % change from reference scenario) with a climate similar to
that at Medicine Hat.

| Variable | HRU | Temperature: 0 °C Precipitation (%) | | | | | Temperature: 2 °C Precipitation (%) | | | | | Temperature: 6 °C Precipitation (%) | | | | |
|---|---|---|---|---|---|---|---|---|---|---|---|---|---|---|---|---|
| | | 0 | −20 | +10 | +20 | +30 | 0 | −20 | +10 | +20 | +30 | 0 | −20 | +10 | +20 | +30 |
| **Annual** | Cultivated | 60 | 46 | 68 | 77 | 84 | 48 | 38 | 52 | 59 | 65 | 44 | 34 | 49 | 54 | 59 |
| **peak SWE** | Grassland | 67 | 50 | 74 | 84 | 92 | 50 | 40 | 56 | 64 | 70 | 46 | 35 | 51 | 57 | 62 |
| **(mm)** | Shrubland | 71 | 52 | 80 | 92 | 101 | 51 | 40 | 58 | 67 | 75 | 46 | 35 | 52 | 59 | 65 |
| | Wetland | 133 | 93 | 154 | 178 | 197 | 82 | 54 | 98 | 120 | 138 | 65 | 44 | 77 | 94 | 111 |
| **Annual runoff (mm)** | | 9 | 4 | 12 | 15 | 19 | 6 | 3 | 8 | 11 | 15 | 5 | 2 | 6 | 9 | 12 |

The date of annual peak SWE advances as annual air temperature warms (Figure 7). The

maximum advance in the date of the annual SWE peak (with no change in precipitation) is about

40 days. The date of peak SWE is more sensitive for climates typical of western Alberta (e.g.

Calgary) than those further east (e.g. Regina, Brandon). For example, an increase of 2°C is

needed to advance peak SWE date by 20 days near Calgary while a similar change requires 4°C

in Saskatchewan or Manitoba. This could be due to the colder initial temperatures further east

and north (Table 2). The nearly horizontal isochrones in Figure 7 suggest that the peak SWE

date is less a function of potential changes in precipitation amount and more a function of

anticipated temperature increases.



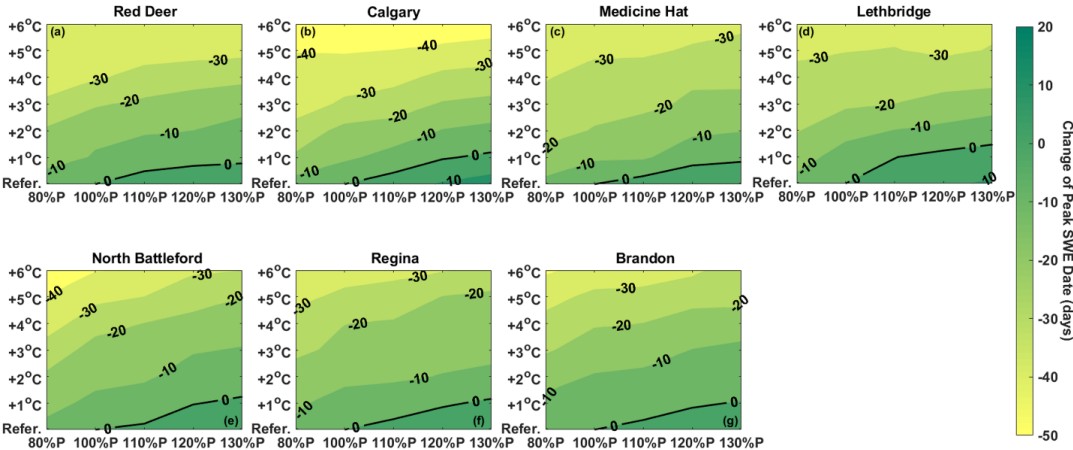

Figure 7: Change in peak annual snow water equivalent date at the wetland HRU under warming and changes in precipitation for simulations using the perturbed 30 year climate data. Negative signs in peak SWE date plot represent advance in time and positive signs denote delay in date of peak SWE accumulation.


Warming is associated with an unequivocal decrease in the amount of snowfall, as the ratio of rain to annual precipitation can increase by more than 10% with warming of 6°C (Figure 8). Greater absolute decreases in wetland HRU SWE with warming were simulated using climate from the eastern and northern edges of the class (i.e., Brandon, North Battleford and Regina)

(Figure 8). Under warming of 6°C annual peak SWE in the wetland HRU decreased by 100 mm in a climate such as Medicine Hat's and as much as 220 mm in a climate such as Brandon's. The absolute differences are due to the drier climates of the western locations (Table 2) that produce smaller baseline annual peak SWE (157 mm at Medicine Hat vs. 409 mm at Brandon). The dry climates also tended to be warmer, with a rain ratio more sensitive to warming (e.g., Lethbridge

vs. North Battleford, Figure 8). This results in a higher rate of loss in SWE in a climate such as southern Alberta's (11% loss in annual peak SWE per °C) compared to climates from the eastern portions (5% loss in annual peak SWE per °C) (Figure 9).

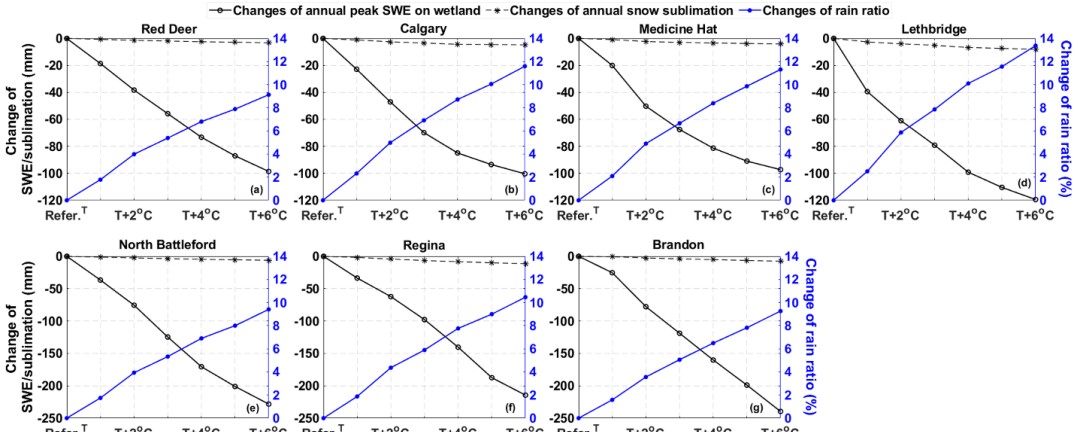

Figure 8:  Changes in mean annual peak snow water equivalent (mm) in a wetland HRU, annual
blowing snow sublimation (mm) and the change in the mean ratio of rain in annual precipitation
(%). Change of rain ratio was calculated by rain ratio of scenario (%) minus the rain ratio of
reference (%).

The changes in rainfall ratio illustrated in Figure 8 are evident when model simulations are

extrapolated across the HEG class (Figure 9).  In much of the class, the annual maximum spring

SWE declines by more than 7% for each °C of warming.  The most sensitive regions are in

western Alberta. When temperature change is held constant in the model, annual maximum SWE

changes almost proportionally with precipitation (Table 4).  When the effects of temperature and

increasing precipitation are combined, there is a trend from east to west of larger decreases in

annual maximum SWE (Figure 9) with a warming and wetting climate.  The predicted increase

of 30% in annual precipitation does not offset the impact of increases in temperature on SWE,

but the maximum annual SWE reductions per degree slow and range from 8% to 1%.  The most

sensitive regions become concentrated in western Alberta as these become relatively drier.

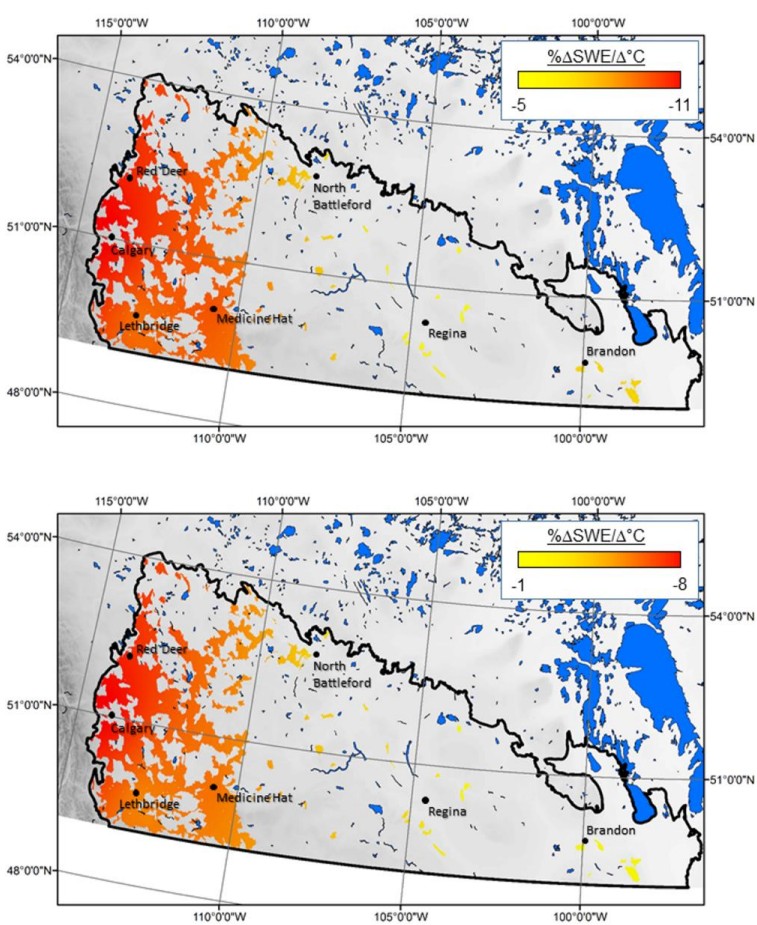


Figure 9: Sensitivity of annual maximum snow water equivalent (SWE) in the grassland HRU to warming, assuming no change in precipitation (top), and a 30% increase in mean annual precipitation (bottom).

*Streamflow*

Absolute decreases in runoff in the climates of Red Deer, Calgary and Lethbridge are all similar

under a warmer climate (Figure 10). Similar, but larger absolute decreases were simulated for the

colder climates of Brandon, Regina and North Battleford.  These colder sites do not see as large

an increase in the rainfall ratio. (Figure 10).  The smallest absolute change in annual runoff (7





mm) was found for areas with a Medicine Hat climate; however, this represents a 78% decrease

in runoff, as baseline runoff (9 mm; Table 4) was the smallest of the seven climates investigated.

Two reference conditions can result in lower relative change in runoff with warming, either a dry

or cold climate.  This is evident in Figure 10. For each °C of warming, streamflow from HEG

basins with western Alberta climates decrease 15%, but in those with climates from the east and

north, this impact diminishes to 7% per °C of warming.  Under a warmer and wetter climate (6°C

warming and 30% increase in annual precipitation) runoff in western portions still experience

decreases, but runoff in climates such as Brandon's increase and remain the same in climates

such as Saskatchewan's (Figure 11).

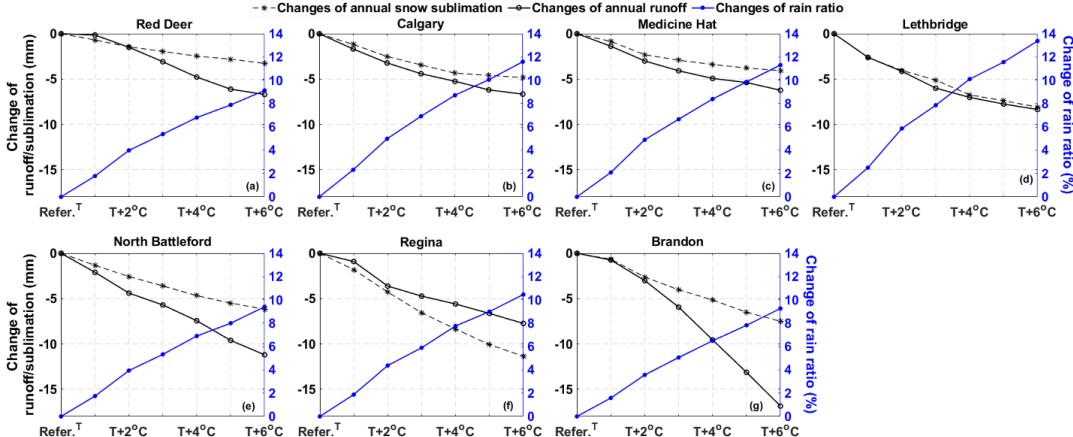

Figure 10:  Change in runoff (mm), annual blowing snow sublimation (mm) and change in the
ratio of rain in annual precipitation (%) for each climate (annual blowing snow sublimation and
ratio of rain as in Figure 8).



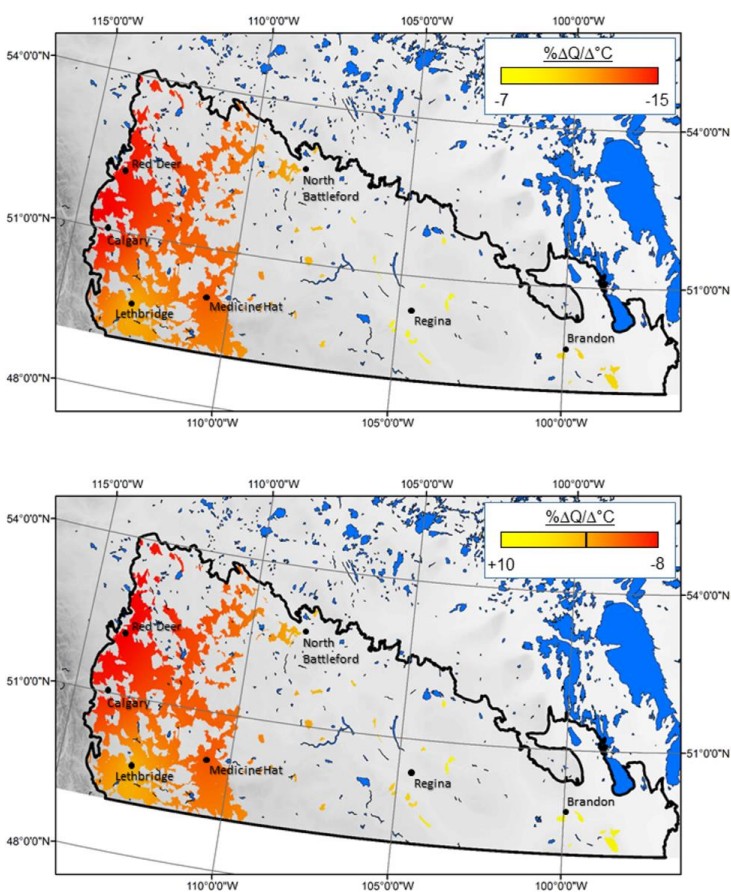

Figure 11: Sensitivity of annual runoff from HEG basins to warming assuming no change in precipitation (top), and a 30% increase in mean annual precipitation (bottom). Note that the black bar on the legend for the bottom panel denotes the colour at which there is no change.

**Discussion**

It is difficult to glean from observations alone how Canadian Prairie basins respond to change

(Ehsanzadeh, 2016), because neither climate nor land-use have remained stationary over the last

half century. The value of using a virtual basin model is the ability to isolate the influence of

individual drivers such as climate on hydrological responses. The model results imply that the


effects of temperature and precipitation changes will differ amongst HRUs. Increases in

precipitation may also cancel the direction and slow the magnitude of changes in streamflow

caused by increased temperatures, and these patterns vary across the region.

*Hydrological sensitivity to climate change*

The climate of the Canadian Prairie has been warming since the mid 20th century, with more

variable and greater precipitation (DeBeer et al., 2016; Gan 1998; Millett et al. 2009), which has

altered water availability and flood characteristics (Pomeroy et al., 2009) which are of interest to

water managers and others. The uncertainty in future climate change projections, particularly in

precipitation (Bush and Lemmen, 2019), can be partially mitigated by the type of sensitivity

analysis conducted here, which can provide some information on how hydrological states and

fluxes may respond to alternate future climate conditions and to identify potential thresholds in

hydrological response. On the Canadian Prairie, the sensitivity to climate varies among

landscape components. There is confidence that annual warming of 6°C coupled with increases

in precipitation of 30% for HEG basins will induce detectable changes in spring snow condition,

with snowpacks in cultivated fields being less sensitive than wetlands. This type of change could

happen by the late 21st century. The sensitivity analysis implies that a 6°C warming alone could

result in a 27% decline in annual maximum SWE in cultivated areas and a 51% decline in

depressions (Table 4). Blowing snow redistribution to depressions is particularly sensitive to

warming temperatures (Figure 8). The increase in temperature reduces snow accumulation and,

in turn, the redistribution of blowing snow to sink areas such as depressions. Rasouli et al.

(2014) also documented a non-linear response in annual peak SWE to changes in temperature

and precipitation, albeit in a mountain basin, where even a 30% increase in precipitation could





not compensate for the impacts of a 6°C warming on peak annual SWE. The reduction in

depression SWE has important implications for groundwater recharge as depressions provide

recharge at volumes disproportionate to their area on the landscape (Hayashi and van der Kamp,

2005). Given that wetland depressions of the region are well known as biodiversity hotspots

(Mantyka-Pringle et al. 2019), these changes could have cascading effects for biophysical

systems.

Warming in the spring and reduced winter snow accumulation advance the timing of annual peak

SWE. With an increase of 6°C, the timing of annual peak SWE advances by nearly six weeks.

This lengthening of the snow free season enhances the opportunity for evaporative losses. The

consequence for streamflow is a 44% decline in annual volume in a drier HEG climate such as

Medicine Hat's (Table 4). Year-to-year variation may be more extreme than suggested by the

model because streamflow variability in the virtual basin model simulations is less sensitive to

inter-annual variation in climate than streamflow records for gauged basins within the

classification indicate. The results imply that changes in streamflow due to warming can be

offset by rising precipitation. Warming of this magnitude has occurred in the last 60 years in

southern Alberta (DeBeer et al., 2016) and the influence can be seen in the smallest simulated

SWE values typically appearing later in the simulation period (Figure 6). Whitfield et al. (2020)

found that basins that align with the HEG class are already responding to this kind of climate

change, exhibiting earlier and smaller runoffs, implying the area is becoming drier. This is

expected to continue with a further 2°C increase from current conditions (i.e., ~4 - 5°C from the

mid 20[th] century) predicted by 2040 (Zhang et al., 2019). The value in understanding the

dynamics of combined impacts of temperature and precipitation changes on SWE and



streamflow is that it can highlight conditions that require more attention for mitigation. The diversity in streamflow response across the class, with greater decreases in streamflow simulated in the west and possibly increases in the east (Figure 11) may require a variety of regionally specific policies and practices to be implemented across the region. This information could be

helpful to water managers and policymakers in the region, including informing how small dams are operated (Muzik, 2001), and wetlands are managed (Wilson et al., 2019).

The sensitivity of hydrological regimes documented here is similar to those of other studies conducted in the HEG classification and other nearby Prairie basins. Fang and Pomeroy (2007)

and Pomeroy et al. (2007) attempted to determine the sensitivities of snow accumulation and runoff to drought at Bad Lake, Saskatchewan. They adjusted initial soil moistures, vegetation heights and winter temperature and precipitation in ways that were informed by previous droughts and by the Global Climate Model predictions of the time. In their work, simulated snow cover duration decreased by 40 days given a 1°C winter warming and a 15% increase in

winter precipitation, which may suggest greater sensitivity to changing winter climate than our results under similar climate perturbations (single digit advancement of timing and modest decreases in amount of peak annual SWE; Figure 8). Muzik (2001) altered precipitation in a simulation of the Little Red Deer River, Alberta, and also found disproportionate changes in streamflow. Using downscaled values from Providing Regional Climates for Impacts Studies

(PRECIS) and the Canadian Regional Climate Model (CRCM) and stochastic weather generator data to force an application of the Soil Water Assessment Tool (SWAT) Model to a Prairie Pothole basin in Saskatchewan, Zhang et al. (2011) found winter runoff increases to be a function of how frequently the air temperature exceeds 0°C. Simulations of Beaver Creek,



Alberta to quantify climate change impacts on snow accumulation and streamflow indicated that

changes in precipitation of more than 10% are needed to alter annual runoff (Forbes et al., 2011).

Rasouli et al. (2014) suggest that in mountainous areas of Alberta, runoff is more sensitive to

changes in precipitation than temperature, however our results in the Canadian Prairies imply a

more dynamic system.  When there is warming and drying, runoff is sensitive to both

precipitation and temperature, but as conditions become wetter, the Prairie system becomes less

sensitive to temperature.   This is not necessarily due to changes in the snowpack, which

responds in a more linear manner (Figure 9), but perhaps due to higher amounts of rainfall

relative to the storage capacity of the landscape.

**Conclusions**

Virtual experiments have proven to be suitable to diagnose the hydrological response of basins in

this landscape. This work introduces a modelling approach that can be used to generate new

knowledge of hydrological behaviours at a regional scale under different boundary conditions.

Where previous studies have focussed on the sensitivity of individual basins in cold regions, and

conclusions about wider applicability were made by assuming the basins to be representative,

this approach is meant to provide a methodology to assess how regional hydrology may respond

to change by modelling a prototypical virtual basin that represents one type of basin in a region.

The results of this study can be used, in part, to address demands for regional-scale information.

Such information is required to develop informed policies for climate change mitigation, but also

more broadly for water resources management in this landscape where land use is predominantly

for intensive agriculture.  However, there are some limitations as the model assumes a

homogenous basin with an area of 100 km$^2$. It also does not account for second order changes





such as how agricultural practices will adapt in response to a warmer and wetter environment

over time and the role these landscape changes play in influencing hydrological processes.

Outputs of virtual experiments are less useful in predicting exact future system states than in

specifying how alternative climate possibilities would alter hydrological behaviour. Therefore,

the model output should be interpreted with some caution as the approach is less useful in

predicting exact future system states than in specifying how alternative choices could alter the

tendency to move towards each of those conditions.

The simulations provide novel insights into the interactions that influence the Prairie

hydrological response to different climates. The experimental design was successful in

simulating the sensitivity of a basin class in the Canadian Prairie to expected climate change.

The scenario results show that the hydrology of HEG basins in the Canadian Prairies is highly

sensitive to changes in climate. What should be considered is that the changes simulated by the

virtual basin model are already underway.  For instance, the mean annual temperature at

Medicine Hat for the baseline simulation period (1960 – 2006) was 5.5°C.  The region is

expected to warm 2°C above the 1976 – 2005 climate normal (6.1°C) to 8.1°C by 2040 (Zhang et

al., 2019).  It is reasonable to expect that the 2°C (i.e., 7.5°C) warming scenario presented here

will be the normal by the end of this decade.  Water managers should expect 38% less snow in

depressions and 20% less snow in cultivated fields in the spring, and 33% less annual runoff.  Of

course, this depends on how wet the next ten years are.  Simulations imply that a 10-20%

increase in average annual precipitation would be needed to offset the warming.  These are

profound differences in the response of the hydrological regime of the region to a warmer and



wetter climate. Water managers and agricultural producers need to consider carefully how they

can adapt their practices in light of such changes.

**Data Availability:** All model forcing datasets used in this research are publicly available and can be accessed via the references and links provided. The virtual basin model outputs are available from the authors by request.

**Author contributions:** CS and CJW conceived the study. ZH, BM and KRS lead the modelling effort and data analysis. JDW lead the catchment classification. All authors contributed to writing.

**Competing interests:** The authors declare that they have no conflict of interest.

**Acknowledgments:** This research was funded by the Canada First Research Excellence Fund's

Global Water Futures programme.





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
