# Peer review of "Assessing hydrological sensitivity of grassland basins in the Canadian Prairies to climate using a basin classification–based virtual modelling approach"

_Hydrology and Earth System Sciences, 2021_

## Author Comment (AC1)

**Response to Reviewer #1**

*Summary*

This study aims to demonstrate the utility of a hybrid classification and virtual modelling approach for assessing the sensitivity of a portion of Canadian Prairie catchments to climate. They first developed a class-based virtual basin model for a portion of the Canadian Prairie and then used the virtual basin to explore the sensitivity of hydrological response to climate.

*Assessment*

I enjoyed reading this manuscript. Class-based virtual basin modeling approach is an innovative tool to roughly identify regional-scale landscape vulnerability. I find the manuscript provides some interesting material for readers of HESS. This seems an appropriate topic to further generalize and extrapolate basins hydrologic functions. However, I think this paper needs major revisions and clarification on the rational of the study, and manuscript presentation and methodology. After these issues are resolved, I believe this paper could be a nice addition to HESS.

Thanks for the comments and encouragement to continue with the review process.

*Major Points:*

This paper uses only one class of basin developed in the classification approach published by the authors before. This class includes a portion of Canadian PPR. Almost up to the middle of the paper this point is not very clear.

As an example, in line 109-111 the objective of the paper was explained as: "This paper aims to demonstrate the utility of a basin classification–based virtual modelling approach for assessing the sensitivity of Canadian Prairie catchments to climate". But they only explored one class out of 7 classes of catchments within Canadian PPR. I suggest that they explain only one class throughout the paper and focus the text to this class only.

We are explicit that we focus on the High Elevation Grasslands class early in the paper (Lines 143 – 155) We can address this comment by better weaving in "the High Elevation Grassland virtual basin" into the text more, rather than only say "the virtual basin".

The point that the classification approach they developed in the previous published work is insensitive to inclusion of climatic features could show the inability of classification approach in categorizing basins based in their functional similarities. The timing, rate, frequency and type of precipitation could be the most dominant factors controlling how a landscape functions irrespective of their physical characteristics.

For evidence that the classification approach was not insensitive (but not for inclusion in the paper) below we list the areas of each catchment class, and their differences. Some classes were not much different (e.g., Sloped incised and High Elevation Grasslands), but some changed by 20% or more (e.g., Interior Grasslands and Southern Manitoba). This provides some comfort that the classification approach was able to capture nuances that differentiate functionality. Climate can be a dominant factor controlling hydrology, agreed. However, we chose not to include climate in this classification as we would be varying and manipulating the climate to explore the variability of current hydrological

response and sensitivity to future climates in the region.  We will remove the sentence on Line 139 because it is misleading, as yes, the two classifications were similar, but different in several ways. Furthermore, we will rewrite this content at the end of this paragraph to be "This was done because including climate could bias the results of the climate scenarios by "hard wiring" some functionality into the model parameters."

| | This papers delineation (km$^2$) | Wolfe et al. 2019 (km$^2$) | Fractional difference |
|---|---|---|---|
| High Elevation Grasslands | 79667.52 | 75176.03 | 0.94 |
| Interior Grasslands | 47211.54 | 57812.5 | 1.22 |
| Major River Valleys | 34533.41 | 39892.18 | 1.15 |
| Pothole Glaciolacustrine | 77844.3 | 70399.25 | 0.90 |
| Pothole Till | 120881.2 | 106541.9 | 0.88 |
| Sloped Incised | 35388.6 | 34432.65 | 0.97 |
| Southern Manitoba | 21149.17 | 32421.21 | 1.53 |

The major assumption of the paper is that the authors previously published classification approach can classify landscapes based on (dis)similarity in their hydrologic functions. However, Oudin et al. [2010] clearly demonstrated that physically similar catchments, with similar physical features, are not truly functionally similar within the context of basin classification. As they used physical features to classify catchments, it is not clear if their classification is functional.

Further to our points above, we can cite a new paper that used hydrological response to classify watersheds on the Prairie (Whitfield et al., 2021) and how it qualitatively aligns well with our classification.  This paper's classification did include streamflow in its calculations and had the purpose of designing the virtual basin model structure and for its parameter selection.  We will also reference Wolfe et al. (2019) which showed that the approach to classification has the capability to be functional.

 If one of the objective of the paper is to demonstrate the utility of a new regionalization approach (e.g., Class-based virtual basin modeling approach) below points should be clarified and discussed in the introduction, method, and discussion sections:

The rational for needing such approach: In which way the approach works better than other regionalization approaches in the literature (see Blöschl et al. [2014] for details of different regionalization approach).

We can explain this further in the first section in methodology as we describe the framework.  In particular, we will include content such as "The low density of gauged non-regulated streams in the region make extrapolation of observed streamflow data highly uncertain."

How computationally efficient is the method? Is it faster than, for example, the approach suggested in Knoben et al. [2018] that only used three functional indices to globally regionalize basins streamflow signatures. If not, why we have to use the proposed method? Does it respect functional behaviour of basins stronger than other methods? How accurate is their method compared to the other regionalization schemes? The paper shows some graphs related to the accuracy of the method. But there is no quantitative and/or qualitative comparison with other methods.

We interpret this comment as not so much about the modelling as much as it is the classification, which is not the point of this paper.  In our defense, Knoben et al.'s paper used three indices, but applied them across the globe.  It is arguable that much of the Prairie could not be differentiated using those three indices (aridity, the seasonal range of water-versus-energy availability, and the fraction of snowfall), as they are similar enough across the Canadian Prairie, relative to the globe, and this is reflected in the figures within Knoben et al..  There are important differences in small basins across the Canadian Prairies, differences due to topography, soils, depressional storage, runoff connectivity within the basin and groundwater connectivity to surface hydrology.  This classification was intended to elucidate and define these differences for the purpose of hydrological model design and parameterisation.

We can include a sentences such as "A more thorough detailed approach is needed within this region than methods such as Knoben et al. (2018) which are too coarse to differentiate functional differences.  This classification was intended to elucidate and define these differences for the purpose of hydrological model design and parameterisation."

There is a dearth of literature on the impacts of climate change and wetland drainage on streamflows of PPR watersheds in both USA and Canada. The authors mostly referred to their previous papers in this regard. In both introduction and discussion, other groups' works must be acknowledged. It should be clarified, how their findings are (in)consistent with other works.

Will we expand on this more in the Introduction (framing the problem) and Discussion (context of other studies).  We thought we were keeping the self-citations down, but we will expand on current content in Lines 568 – 592 that cites other groups' work papers by including papers that Golden and Johnson wrote in the region in nearby North Dakota.

*Minor Points:*

Line 44: How long does it take to run one model. How long would it have taken to run a model for every catchment in the database or every catchment in the cluster? No talk of efficacy of virtual experiments in discussion, yet it seems like one of the central purposes of your paper in the introduction.

For each 46 year run (1960 to 2006), the model takes around 7 minutes on a PC with CPU of Intel Core i7-8650U.  One reason we could not run the model for each of the 796 catchments in the HEG class is that we could not find a decent gridded climate dataset (we tried three).  So, we are currently running the model based on the availability of station observations. This resulted in the seven AHCCD stations. With 35 climate scenarios run for seven stations, it took 7 hours to complete all the simulations. We can add these details to the methodology and add a paragraph about efficiency to the discussion.

Line 139: Why would there be bias?

We can add content to address this question such as: "The influence of recent climate, historical climate change and how this interacts with Prairie biogeophysical features and drainage to influence basin hydrology behaviour has been recently explored by Whitfield et al. (2021).  They showed the coherence between climate effects and biogeophysical and drainage effects on basin classification.  Therefore, bias could be introduced from a classification that includes climate when applying future climates to force the virtual basin model.  For instance, if a current climate classification resulted in a virtual basin class that precluded semi-arid basins, this would become non-functional with the projected increase in precipitation and temperature for the region which could change the area to semi-arid.  By excluding

the direct impact of climate in the classification and restricting this to topography, biogeophysical features and drainage systems then the new climate of a set of basins can be more fully explored."

Whitfield et al., The spatial extent of hydrological and landscape changes across the mountains and prairies of Canada in the Mackenzie and Nelson River Basins based on data from a warm-season time window, Hydrology and Earth System Sciences, 25, (2021), 2513:2541, doi: 10.5194/hess-25-2513-2021.

Lines 209-210: Why is the maximum threshold less than minimum?

We missed mentioning the minimum is for 100% rainfall. We can fix this by editing the sentence to "… for 100% snow and a minimum temperature threshold of +2°C for 100% rainfall and distributes ….."

Line 193: (Shook et al., 2013) lots of other places where comma is missing as well

We will fix these.

Line 283: Why don't you look at cluster center (compare median catchment with virtual model)

We can add a sentence such as; "The cluster median was not used for comparison, as the range as represented by snow on the ground (Figure 4) and streamflow (Figure 5) better demonstrates that model simulations are within the envelope of plausible hydrological functioning rather than comparable to an individual basin within the class."

Line 434: To me advance in max snow depth date implies further in time, i.e., closer to summer. Rephrase.

We believe that in many studies looking at the day of year of maximum snow water equivalent, or snowmelt or snow free dates, advance implies it is arriving earlier. We can rephrase to "The date of annual SWE peak (with no change in precipitation) arrives 40 days earlier."

Line 517: should it be "temperatures , but these patterns"?

Respectfully, no. The patterns that vary across the region reflect how precipitation may cancel the direction and slow the magnitude of streamflow changes caused by increased temperatures.

Blöschl, G., M. Sivapalan, M. Wagener, A. Viglione, and H. Savenije (2014), Runoff Prediction in Ungauged Basins, Cambridge University Press.

Knoben, W. J. M., R. A. Woods, and J. E. Freer (2018), A Quantitative Hydrological Climate Classification Evaluated With Independent Streamflow Data, Water Resources Research, 54(7), 5088-5109.

Oudin, L., A. Kay, V. Andréassian, and C. Perrin (2010), Are seemingly physically similar catchments truly hydrologically similar?, Water Resources Research, 46(11).

---

## Author Comment (AC2)

**Response to Reviewer #2**

Overall, Spence et al. did a fantastic job presenting the development and application of a Region-wide hydrological model used to test the sensitivity of Prairie water budgets to changes in precipitation and temperature that are expected in the future. The manuscript is extremely well written and easy to follow with ample justification and explanation of the limitations of their modeling approach. This study also sets the stage for future modeling studies as well.

In the future it would be very interesting to compare seasonality changes with actual GCM outputs rather than using the delta method to enact changes in precipitation and temperature on historical records. These seasonality changes will especially be important for your peak SWE and runoff estimates. Also, in the discussion you allude to the fact that land use and land cover has also been changing. It would be very valuable to use this modeling approach to quantify climate and land-use change synergies and simultaneous combined impacts on hydrology in the future.

We agree with the reviewer. This paper was written as a proof of concept to show that we could use this approach for future sensitivity studies to both land use and climate changes, and not just one at a time. We agree that GCM outputs could be viable driving data, but we have struggled to find good regional climate model data that we have confidence in. We recognize that the delta method is somewhat simple, but it remained the least problematic approach. We can add content discussing this in relevant sections. Also, see our comment regarding gridded climate data in our responses to Reviewer #1.

L115 remove "land management scenarios" as you only looked at responses to changes in precipitation and temperature across one ecoregion.

Thank you for the suggestion but we feel that it is important to mention land management scenarios here as we hope to apply the framework to these in the future, as this reviewer suggests above.

L192 Please explain your wetland complex configuration in a bit more detail. Where does the 46 wetlands come from? In L344-347 the results are presented in the context of wetland density and commenting on the relatively dense drainage networks coupled with small wetland densities. How are the wetland densities dealt with in the model? See the two citations below for very recent evidence that including an areal estimate of wetland depressions within your HRU can improve streamflow discharge estimates like those in Figure 5.

We can do this by citing Pomeroy et al., 2009 and explaining how the configuration was conceptualized with a few sentences at the end of this paragraph.

L366 Please quantify "reasonable simulations of streamflow" using some comparison of means.

We have the data we used to create Figure 5 and will now provide numbers for the four climate stations and include a sentence such as "As with SWE, the CRHM HEG virtual basin model produces reasonable simulations of streamflow. Using Red Deer climate conditions, simulated and observed mean annual streamflow was X and Y. Similarly these values were A, B C, D and E and F for Calgary, Lethbridge and Medicine Hat climate conditions, respectively."

L368 see previous comment regarding the use of "good agreement"

Please see our response above.

In your data availability statement please provide links to the data

We can do this.  We have uploaded climate data inputs, model outputs and model parameter files to the Federated Research Data Repository. The data are still under the verification of a curator. Once verified, the data will be publicly available.  We expect this to be before the paper is published, but we already have the doi's that will be used, which we will include.

Rajib, A., Golden, H. E., Lane, C. R., & Wu, Q. (2020). Surface depression and wetland water storage improves major river basin hydrologic predictions. Water Resources Research, 56, e2019WR026561. https://doi.org/10.1029/2019WR026561

Golden, H. E., Lane, C. R., Rajib, A., & Wu, Q. (2021). Improving global flood and drought predictions: integrating non-floodplain wetlands into watershed hydrologic models. Environmental Research Letters.

---

## Author Response (AR1)

**Reviewer #1**

*Summary*

This study aims to demonstrate the utility of a hybrid classification and virtual modelling approach for assessing the sensitivity of a portion of Canadian Prairie catchments to climate. They first developed a class-based virtual basin model for a portion of the Canadian Prairie and then used the virtual basin to explore the sensitivity of hydrological response to climate.

*Assessment*

I enjoyed reading this manuscript. Class-based virtual basin modeling approach is an innovative tool to roughly identify regional-scale landscape vulnerability. I find the manuscript provides some interesting material for readers of HESS. This seems an appropriate topic to further generalize and extrapolate basins hydrologic functions. However, I think this paper needs major revisions and clarification on the rational of the study, and manuscript presentation and methodology. After these issues are resolved, I believe this paper could be a nice addition to HESS.

**Thanks for the comments and encouragement to continue with the review process. We address each comment in turn. Any line numbers we refer to are those in the marked up version of the manuscript.**

*Major Points:*

This paper uses only one class of basin developed in the classification approach published by the authors before. This class includes a portion of Canadian PPR. Almost up to the middle of the paper this point is not very clear.

As an example, in line 109-111 the objective of the paper was explained as: "This paper aims to demonstrate the utility of a basin classification–based virtual modelling approach for assessing the sensitivity of Canadian Prairie catchments to climate". But they only explored one class out of 7 classes of catchments within Canadian PPR. I suggest that they explain only one class throughout the paper and focus the text to this class only.

**We are explicit that we focus on the High Elevation Grasslands class early in the paper. We addressed this comment by better weaving in "the High Elevation Grassland virtual basin" into the text more, rather than only say "the virtual basin". We added content to address this at Lines 113, 271, 285, 323. We tried to be specific that this exercise was for only one class of Canadian Prairie catchments. In the model application section (beginning Line 265) we added come content to make sure it was clear that the focus was on the HEG (High elevation grasslands) class. We also added content (Line 616) in the conclusions to make sure the reader knows our conclusions are for the HEG class.**

The point that the classification approach they developed in the previous published work is insensitive to inclusion of climatic features could show the inability of classification approach in categorizing basins based in their functional similarities. The timing, rate, frequency and type of precipitation could be the most dominant factors controlling how a landscape functions irrespective of their physical characteristics.

**For evidence that the classification approach was not insensitive (but not for inclusion in the paper) below we list the areas of each catchment class with climate included (Wolfe et al., 2019) and**

**excluded (this paper), and their differences. Some classes were not much different (e.g., Sloped incised and High Elevation Grasslands), but some changed by 20% or more (e.g., Interior Grasslands and Southern Manitoba). This provides some comfort that the classification approach was able to capture nuances that differentiate functionality. Climate can be a dominant factor controlling hydrology, agreed. However, we chose not to include climate in this classification as we would be varying and manipulating the climate to explore the variability of current hydrological response and sensitivity to future climates in the region. We removed the sentence at the end of this paragraph (Line 138) because it is misleading, as yes, the two classifications were similar, but different in several ways. Furthermore, we rewrote the content at the end of this paragraph to be "This was done because including climate could bias the results of the climate scenarios by "hard wiring" some functionality into the model parameters." (Lines 146-162)**

|  | This papers delineation (km$^2$) | Wolfe et al. 2019 (km$^2$) | Fractional difference |
|---|---|---|---|
| High Elevation Grasslands | 79667.52 | 75176.03 | 0.94 |
| Interior Grasslands | 47211.54 | 57812.5 | 1.22 |
| Major River Valleys | 34533.41 | 39892.18 | 1.15 |
| Pothole Glaciolacustrine | 77844.3 | 70399.25 | 0.90 |
| Pothole Till | 120881.2 | 106541.9 | 0.88 |
| Sloped Incised | 35388.6 | 34432.65 | 0.97 |
| Southern Manitoba | 21149.17 | 32421.21 | 1.53 |

The major assumption of the paper is that the authors previously published classification approach can classify landscapes based on (dis)similarity in their hydrologic functions. However, Oudin et al. [2010] clearly demonstrated that physically similar catchments, with similar physical features, are not truly functionally similar within the context of basin classification. As they used physical features to classify catchments, it is not clear if their classification is functional.

**Further to our points above, we cited two papers that clustered catchments by streamflow patterns (Ellis and Gray, 1966 and Whitfield et al., 2021) and how they qualitatively align well with our classification (Line 347). We added content to explain this paper's classification did include streamflow in its calculations so it should capture functionality (new content on Lines 152-162).**

If one of the objective of the paper is to demonstrate the utility of a new regionalization approach (e.g., Class-based virtual basin modeling approach) below points should be clarified and discussed in the introduction, method, and discussion sections:

The rational for needing such approach: In which way the approach works better than other regionalization approaches in the literature (see Blöschl et al. [2014] for details of different regionalization approach).

**We addressed this comment by explaining further in the first section in methodology when we describe the framework (Line 123). In particular, we included content such as "The low density of gauged non-regulated streams in the region make extrapolation of observed streamflow data highly uncertain."**

How computationally efficient is the method? Is it faster than, for example, the approach suggested in Knoben et al. [2018] that only used three functional indices to globally regionalize basins streamflow

signatures. If not, why we have to use the proposed method? Does it respect functional behaviour of basins stronger than other methods? How accurate is their method compared to the other regionalization schemes? The paper shows some graphs related to the accuracy of the method. But there is no quantitative and/or qualitative comparison with other methods.

**We interpret this comment as not so much about the modelling as much as it is the classification, which is not the point of this paper. In our defense, Knoben et al.'s paper used three indices, but applied them across the globe. It is arguable that much of the Prairie could not be differentiated using those three indices (aridity, the seasonal range of water-versus-energy availability, and the fraction of snowfall), as they are similar enough across the Canadian Prairie, relative to the globe, and this is reflected in the figures within Knoben et al.. There are important differences in small basins across the Canadian Prairies, differences due to topography, soils, depressional storage, runoff connectivity within the basin and groundwater connectivity to surface hydrology. This classification was intended to elucidate and define these differences for the purpose of hydrological model design and parameterisation.**

**We included a sentence on Line 134: "A more thorough detailed approach is needed within this region because methods such as Knoben et al. (2018) are too coarse to differentiate functional differences over this region. This classification was intended to elucidate and define these differences for the purpose of hydrological model design and parameterization."**

There is a dearth of literature on the impacts of climate change and wetland drainage on streamflows of PPR watersheds in both USA and Canada. The authors mostly referred to their previous papers in this regard. In both introduction and discussion, other groups' works must be acknowledged. It should be clarified, how their findings are (in)consistent with other works.

**We thought we were keeping the self-citations down, and removed more for this version as some were repetitive. We expanded on current content that cites other groups' work papers by including papers from Johnson who wrote about nearby North Dakota and Muhammad who wrote about a catchment in Manitoba (Line 76, 559, 582).**

*Minor Points:*

Line 44: How long does it take to run one model. How long would it have taken to run a model for every catchment in the database or every catchment in the cluster? No talk of efficacy of virtual experiments in discussion, yet it seems like one of the central purposes of your paper in the introduction.

**We added this details to the methodology (Line 335): "For each 46 year run (1960 to 2006), the model takes around 7 minutes on a PC with CPU of Intel Core i7-8650U. One reason we could not run the model for each of the 796 catchments in the HEG class is that we could not find a decent gridded climate dataset (we tried three). So, we are currently running the model based on the availability of station observations. This resulted in the seven AHCCD stations. With 35 climate scenarios run for seven stations, it took 7 hours to complete all the simulations."**

**We added this content at the beginning of the discussion (Line 499): "The value of using a classification-based virtual basin approach is that it can be more efficient than running models over each catchment in the region. Running the model with one parameterization for a prototypical basin seven times for each climate record plus the 35 climate scenarios took seven hours. The subsequent**

**kriging added another two hours for a total of nine hours. If there had had been reasonable climate data and all 796 basins in the HEG class were run, the simulations would have taken 33 days." We also added this content at Line 504: "A value of using a virtual basin model approach rather than statistically extrapolating observed streamflow records and subsequently attempt to disentangle controls on streamflow is the ability to isolate the influence of individual drivers such as climate on hydrological responses." And moved a sentence from the beginning of this paragraph to better align the thoughts.**

Line 139: Why would there be bias?

**We added content to address this question on Line 152: "The influence of recent climate, historical climate change and how this interacts with Prairie biogeophysical features and drainage to influence basin hydrology behaviour has been recently explored by Whitfield et al. (2021). They showed the coherence between climate effects and biogeophysical and drainage effects on basin classification. Therefore, bias could be introduced from a classification that includes climate when applying future climates to force the virtual basin model. For instance, if a current climate classification resulted in a virtual basin class that precluded semi-arid basins, this would become non-functional with the projected increase in precipitation and temperature for the region which could change the area to semi-arid. By excluding the direct impact of climate in the classification and restricting this to topography, biogeophysical features and drainage systems then the new climate of a set of basins can be more fully explored."**

**Whitfield et al., The spatial extent of hydrological and landscape changes across the mountains and prairies of Canada in the Mackenzie and Nelson River Basins based on data from a warm-season time window, Hydrology and Earth System Sciences, 25, (2021), 2513:2541, doi: 10.5194/hess-25-2513-2021.**

Lines 209-210: Why is the maximum threshold less than minimum?

**We missed mentioning the minimum is for 100% rainfall. We fixed this by editing the sentence (now on Line 222) to "… for 100% snow and a minimum temperature threshold of +2°C for 100% rainfall and distributes …..**

Line 193: (Shook et al., 2013) lots of other places where comma is missing as well

**Fixed.**

Line 283: Why don't you look at cluster center (compare median catchment with virtual model)

**We add a sentence at Line 300 such as; "The cluster median was not used for comparison, as the range as represented by snow on the ground (Figure 4) and streamflow (Figure 5) better demonstrates that model simulations are within the envelope of plausible hydrological functioning rather than comparable to an individual basin within the class."**

Line 434: To me advance in max snow depth date implies further in time, i.e., closer to summer. Rephrase.

**We believe that in many studies looking at the day of year of maximum snow water equivalent, or snowmelt or snow free dates, advance implies it is arriving earlier. We rephrased on Line 541 to "The date of annual SWE peak (with no change in precipitation) arrives 40 days earlier."**

Line 517: should it be "temperatures , but these patterns"?

**Respectfully, no. The patterns that vary across the region reflect how precipitation may cancel the direction and slow the magnitude of streamflow changes caused by increased temperatures.**

Blöschl, G., M. Sivapalan, M. Wagener, A. Viglione, and H. Savenije (2014), Runoff Prediction in Ungauged Basins, Cambridge University Press.

Knoben, W. J. M., R. A. Woods, and J. E. Freer (2018), A Quantitative Hydrological Climate Classification Evaluated With Independent Streamflow Data, Water Resources Research, 54(7), 5088-5109.

Oudin, L., A. Kay, V. Andréassian, and C. Perrin (2010), Are seemingly physically similar catchments truly hydrologically similar?, Water Resources Research, 46(11).

**Reviewer #2**

Overall, Spence et al. did a fantastic job presenting the development and application of a Region-wide hydrological model used to test the sensitivity of Prairie water budgets to changes in precipitation and temperature that are expected in the future. The manuscript is extremely well written and easy to follow with ample justification and explanation of the limitations of their modeling approach. This study also sets the stage for future modeling studies as well.

In the future it would be very interesting to compare seasonality changes with actual GCM outputs rather than using the delta method to enact changes in precipitation and temperature on historical records. These seasonality changes will especially be important for your peak SWE and runoff estimates. Also, in the discussion you allude to the fact that land use and land cover has also been changing. It would be very valuable to use this modeling approach to quantify climate and land-use change synergies and simultaneous combined impacts on hydrology in the future.

**We agree with the reviewer. This paper was written as a proof of concept to show that we could use this approach for future sensitivity studies to both land use and climate changes, and not just one at a time. We agree that GCM outputs could be viable driving data, but we have struggled to find good regional climate model data that we have confidence in. We recognize that the delta method is somewhat simple, but it remained the least problematic approach. We added content mentioning the difficulty we had finding good gridded climate data (Lines 126, 337). Also, see our comment regarding gridded climate data in our responses to Reviewer #1.**

L115 remove "land management scenarios" as you only looked at responses to changes in precipitation and temperature across one ecoregion.

**Thank you for the suggestion but we feel that it is important to mention land management scenarios here as we hope to apply the framework to these in the future, as this reviewer suggests above.**

L192 Please explain your wetland complex configuration in a bit more detail. Where does the 46 wetlands come from? In L344-347 the results are presented in the context of wetland density and commenting on the relatively dense drainage networks coupled with small wetland densities. How are the wetland densities dealt with in the model? See the two citations below for very recent evidence that including an areal estimate of wetland depressions within your HRU can improve streamflow discharge estimates like those in Figure 5.

**We added these sentences beginning at Line 203. We hope it addresses the reviewer's questions and comments:**

**"The selection of 46 wetlands was based on the work of Shook and Pomeroy (2011) and Shook et al. (2013) who found that this was an optimal number of wetlands to represent crucial hydrological behaviour (e.g., hysteresis) that controls the dynamic area contributing flow downstream. The total wetland area was based on the fractional area from basin classification. The distribution of this area among the 46 wetlands followed a generalized Pareto distribution as it has been used successfully to characterize the distribution of open water areas in similar landscapes in North Dakota and central Saskatchewan (Zhang et al., 2009; Shook et al., 2013) To constrain this distribution the location of the largest wetland among the 46 was based on distance of the largest wetland to outlet calculated as part of the catchment classification by Wolfe et al. (2019)."**

L366 Please quantify "reasonable simulations of streamflow" using some comparison of means.

**The section in which runoff ratios and mean annual hydrographs are compared to observed data has been reorganized to address this comment. The runoff ratio material is now presented first, as the thoughts seemed to flow better that way.**

**We have the data we used to create Figure 5 and now provide a comparison of means for the four climate stations and include a sentence at the end of the section (Line 404). "Simulated mean annual runoff across the HEG class averaged 10 mm, while observed runoff from the Water Survey of Canada gauges (Table 3) averaged 17 mm for the verification period of 1976 - 2006. This low bias aligns with the low runoff ratios and the poor representation of convective rainfall events in the meteorological forcing discussed earlier. Furthermore, the Water Survey of Canada often selects basins with median contributing area fractions that approach 1.0, as these are more efficient at producing streamflow and are easier to gauge (Whitfield et al., 2020). This differs from the contributing area fraction of 0.67 Wolfe et al. (2019) measured as the median across the HEG class and used in the virtual model, and this too could account for some of this low bias."**

L368 see previous comment regarding the use of "good agreement"

**Please see our response above.**

In your data availability statement please provide links to the data

**We can do this. We have uploaded climate data inputs, model outputs and model parameter files to the Federated Research Data Repository. The data are still under the verification of a curator. Once verified, the data will be publicly available. We expect this to be before the paper is published, but we already have the doi's that will be used, which we have now included.**

*Rajib, A., Golden, H. E., Lane, C. R., & Wu, Q. (2020). Surface depression and wetland water storage improves major river basin hydrologic predictions. Water Resources Research, 56, e2019WR026561. https://doi.org/10.1029/2019WR026561*

*Golden, H. E., Lane, C. R., Rajib, A., & Wu, Q. (2021). Improving global flood and drought predictions: integrating non-floodplain wetlands into watershed hydrologic models. Environmental Research Letters.*